**Subject Category:**
Biology (whole organism)

ecology

Africa, Ichneumonidae, ecology, species richness, extensive sampling, Uganda Malaise trapping 2014–2015

**Author for correspondence:**
Tapani Hopkins
e-mail: tapani.e.hopkins@utu.fi

# Extensive sampling reveals the phenology and habitat use of Afrotropical parasitoid wasps (Hymenoptera: Ichneumonidae: Rhyssinae)

Tapani Hopkins[1], Heikki Roininen[2] and Ilari E. Sääksjärvi[1]

[1]Zoological Museum, Biodiversity Unit, FI-20014 University of Turku, Finland
[2]Department of Environmental and Biological Sciences, University of Eastern Finland, Joensuu, Finland

TH, 0000-0002-2256-0098; IES, 0000-0002-8107-5607

Tropical invertebrates, such as the ichneumonids of tropical forests, are poorly known. Here, we report the first results of extensive sampling at Kibale National Park, Uganda, by providing some of the first tropical ecological data for the ichneumonid subfamily Rhyssinae. We sampled ichneumonids with 34 Malaise traps for a year in 10 sites, in habitats ranging from primary forest to farmland. We also gathered weather and vegetation data. The total sampling effort was 373 trap months and we caught 444 rhyssines in six species. We caught the most rhyssines in dry weather, and towards the end of the sampling year. The rhyssines showed a clear preference for decaying logs and for primary forest. We fitted a model which can be used to predict future catches at the site, and draw conclusions on when rhyssines emerge and on their adult lifespan. Sampling extensively gave us a wealth of ecological data on a poorly known parasitoid wasp subfamily. We recommend that future tropical sampling collect ecological data, and that existing data from previous large-scale surveys be used for ecological analyses.

## 1. Introduction

Our planet is inhabited by millions of species, which are distributed unevenly in different latitudes and habitats. The global distribution of this species richness follows regular patterns, but describing these patterns—let alone explaining the mechanisms behind them—is only possible if we know much

more about the species that inhabit our planet. Ideally such knowledge includes data not only on species distributions, but also on the ecology of the species, so that the observed distributions can be explained. Currently, however, biodiversity is so undersampled that most terrestrial animal species have not even been discovered [1]. Tropical arthropods are especially poorly known (e.g. [2,3]).

The parasitoid wasp family Ichneumonidae is ecologically significant and extremely species-rich, but its tropical diversity is very poorly known. Ecologically, ichneumonids play an important role in regulating population densities of other insects and spiders ([4] and e.g. [5–7]), and since they are parasitoids, their numbers and diversity reflect that of their hosts [8–10]. They may number over 100 000 species worldwide, of which only 24 000 have been described [11,12]. Until recently, few species were known from the tropics and the family was assumed to be relatively species-poor there ([13,14] but see [15]). Later, more extensive sampling revealed extremely rich ichneumonid faunas in Costa Rica and Amazonia [16–18]. We currently do not have enough data to say whether ichneumonids follow the general pattern and are the most species-rich in the tropics [15,19] or show an anomalous latitudinal diversity gradient (the earlier assumption, which numerous studies have tried to explain [20–24]). Old World tropical forests could well harbour rich ichneumonid faunas, which have simply not been discovered due to insufficient sampling [25].

Extensive long-term sampling appears to be needed to adequately inventory tropical ichneumonid faunas [18,26]. Not only are many tropical ichneumonids seasonal (e.g. [27,28]), but tropical forests are often a rich mosaic of different habitat types (e.g. [29,30]). Tropical ichneumonid species also appear to have low population densities, or at least are hard to catch in large numbers [15,18,26,31]. Finding out what ichneumonid species inhabit an area, and what their ecology is, thus seems to need more than a mere few months of low intensity sampling. It has been suggested that ichneumonids require at least a year of sampling (to cover all seasons) and a large number of traps which cover different habitat types [18,26,32,33].

Tropical ichneumonid faunas have rarely been extensively sampled, with the few published long-term sampling efforts concentrated in Central and South America. The Costa Rican sampling set the stage for later tropical ichneumonid work and has encompassed at least 1500 Malaise trap months (e.g. over 1200 trap months of which 576 were in the most sampled site [16], about 190 trap months [28]). This was followed by an extensive sampling programme in western Amazonia (185 trap months [18], at least 72 additional trap months [26]). African ichneumonids have been extensively sampled in Hantam Botanical Gardens in South Africa, but the results have not yet been published (258 trap months, S. van Noort 2018, personal communication). More limited assessments have been conducted in Sierra Leone and Uganda [13], Namibia [34], Gabon [35], the Central African Republic, Tanzania, Uganda and South Africa (van Noort 2018, personal communication). We have published results of previous sampling in Uganda, but that study caught too few ichneumonids to yield more than superficial ecological results (231 trap months using traps smaller than the standard size [25]). It is clear that much more extensive sampling still needs to be done in the Afrotropics: currently, only 2097 ichneumonid species are known from sub-Saharan Africa [12,36] out of an estimated 12 100 [36,37].

Given that many species have not even been discovered, it is scarcely surprising that very little is known of the phenology and habitat use of tropical ichneumonids. Most of the existing data stem from the Neotropics, from the extensive Costa Rican and western Amazonian surveys, but even they have yielded surprisingly little published information. In Costa Rica, the parasitoid subfamily Ophioninae (that attacks larvae) was caught in greatest numbers soon after the start of the rains, and the Pimplinae and Rhyssinae (that mostly attack prepupae or pupae) near the end of the rainy season [16]. Later, more detailed results from Costa Rica and Panama indicate that the phenology varies from subfamily to subfamily and between sites in ways that defy simple explanation, but overall rainfall has a clear effect [28]. Apart from these results, only fragmentary information on the phenology of Costa Rican ichneumonids appears to have been published. There is somewhat more information on habitat use, e.g. with subfamilies such as Orthocentrinae seeming to prefer humid shaded habitat [16], but much of this information is anecdotal and lacks detail. The only systematic overview we know of found 1.89 times as many ichneumonids in wet forest as in moist forest, with parasitoids of larvae having different habitat preferences to parasitoids of pupae [28]. In Amazonia, seasonal variation, possibly linked to rainfall, has been observed in ichneumonid subfamily catches [38]. The catches of subfamilies Rhyssinae and Pimplinae have also been compared to vegetation, with the ichneumonid assemblage caught by a trap found to be associated with the surrounding vegetation [9]. For the Afrotropics, there are much less published data (other than anecdotal evidence or data for individual species). A single trap in a Ugandan garden caught most ichneumonids at the end of the rainy season

in May–June, and this peak was matched by a peak in lepidopteran numbers [39]. Also, Gauld & Mitchell [27] observed that the distributions of Ophioninae species matched continental scale vegetation zones such as savannah or lowland forest. Overall, previous tropical results give a very incomplete picture of ichneumonid phenology and habitat use, but suggest that rainfall increases catches, and that ichneumonid assemblages are likely to measurably differ between habitats.

The ecology of the ichneumonid subfamily Rhyssinae is poorly known in the tropics, with practically no data from tropical Africa. In general, rhyssines are held to be parasitoids of wood-boring sawfly or beetle larvae [12,16] that prefer humid lowland forests [16]; dry periods and arid areas appear unsuitable [40,41]. Costa Rican rhyssines seem to prefer lowland forest and may peak at the end of the rainy season ([16, (pp. 31–34)]), and Amazonian rhyssine distributions may show some correlation with vegetation ([9], though the effect is obscured by the practice of lumping Rhyssinae and Pimplinae together). As for the Afrotropics, prior to the present study only 30 rhyssine *individuals* were known from the whole of sub-Saharan Africa ([42], though some further specimens may exist in undetermined museum collections). Ecological analyses are impossible with such sample sizes, and, as far as we know, have not been attempted.

In this and a sister paper, we report the first results of an extensive 1-year sampling of Afrotropical ichneumonids in Kibale National Park, Uganda. In the sister paper, we report the taxonomic results for the subfamily Rhyssinae, including descriptions of two new species [43]. Here, we describe the ecology of this subfamily. Specifically, we asked how the rhyssine species were distributed in space and time, i.e. how ecological variables such as weather and habitat type were associated with the number of rhyssines caught in traps.

# 2. Material and methods

## 2.1. Study site

We placed Malaise traps near the Makerere University Biological Field Station (0°33.75′ N, 30°21.37′ E; approx. 1500 m.a.s.l.) in Kibale National Park (795 km$^2$), Western Uganda (figure 1). The park contains medium altitude moist evergreen forest as well as swamps, grasslands, woodland thickets and colonizing shrubs [44,45], and is surrounded by agricultural land including tea plantations and small farms. It was formerly connected to the forest of the Congo Basin although this connection has deteriorated due to human activity [46]. There are two wet and two dry seasons each year [47].

The study area contains nine sites of varying successional status (figure 1). Four former conifer plantations were clearcut at different times and left to naturally regenerate into forest: R03, R01, R98 and R93 (clearcut during 2002–2004, 2000–2001, 1995–1999 and 1987–1994, respectively). These had regenerated for an average of 12, 14, 17 and 22 years at the time of our study, and have, in earlier work, been referred to as RAC9, RAC11, RAC14 and RAC19 [48]. Three 'disturbed forest' sites were partially logged in 1968–1969 and had regenerated for 46 years: K13, K15 and K14 (ordered from most to least logged). Estimates of the extent of the logging vary (e.g. [45,49,50]), but are approximately 50% basal area reduction for K13 (also treated with the arboricide Finopal DT [48,50]), 40% basal area reduction for K15 [50] and 25% trees destroyed for K14 [45]. Two 'primary forest' sites have experienced minimal disturbance: site K30 was minimally logged before 1970 (2–3 trees km$^{-2}$ [45]), and site K31 has confusingly been referred to both as primary untouched forest (e.g. [48]) and as heavily logged forest [44]. Our traps were located in what we believe to be an unlogged part of K31 (the area called K32 by Olupot [51]).

The above nine sites form a successional gradient from primary forest to recently clearcut plantation (K31, K30, K14, K15, K13, R93, R98, R01, R03). The agricultural land outside the national park can also be seen as a continuation of the gradient, and we included it as a 10th 'site'. This gradient can be divided into successional classes at many different resolutions; in this work, we primarily grouped the sites into the four successional classes of primary forest (K31, K30), disturbed forest (K14, K15, K13), former plantation (R93, R98, R01, R03) and farmland.

It should be noted that the successional sites are far from homogeneous. The disturbed forest sites, for example, were not logged evenly—instead, they are a patchwork of strongly logged and barely touched areas [45]. Even the primary forest is a bewilderingly varied mosaic of different habitat patches, with variation in at least elevation, moisture, soil and vegetation. We were largely unable to capture this variation, but made a distinction between traps placed in swampy primary forest (i.e. forest with waterlogged ground such as a swamp or stream) and other primary forest.

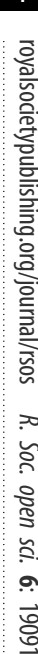

**Figure 1.** Map of the study area in Kibale National Park, Uganda. We placed 34 Malaise traps in primary forest (K31, K30), disturbed forest (K14, K15, K13) and former plantations which had been clearcut 1993–2003 (R93, R98, R01, R03). There were also two traps outside the forest in farmland.

## 2.2. Vegetation and weather

We gathered data on the vegetation surrounding our Malaise traps using two different methods. We listed all the plant species (except trees of at least 5 cm diameter at breast height (dbh)) present in a 5 × 5 m square centred on each trap. We also documented all trees (dbh ≥ 5 cm) in two 50 × 2.5 m transects centred on each trap, with the transects at right angles to each other (see [52] for illustration).

We documented the species, distance from trap and diameter at breast height of living trees—for dead trees, we also estimated the level of decay (1, barely dead; 2, light decay; 3, clearly decaying).

Apart from documenting the vegetation surrounding the trap, we also gathered other background data such as canopy openness and the orientation of the trap and transects. We measured canopy openness (% sky visible from each trap) with a spherical densiometer. We measured the orientation of the trap and the vegetation transects by measuring the direction (in tens of degrees from north) in which the trap head was pointing with a GPS. The vegetation transects were oriented so that one was parallel to the trap orientation, the other at right angles to it.

Weather data were provided by the Makerere University Biological Field Station. The data consisted of daily maximum and minimum temperatures and daily rainfall, which were measured at the field station. The data spanned almost the entire sampling year, 1 August 2014–4 September 2015. We added an estimate of when wet and dry seasons generally occur [53] to these data [52].

## 2.3. Malaise traps and samples

We used 34 Malaise traps to collect flying insects during the sampling year 8 September 2014–14 September 2015. The traps were of a standard size ('Marris House Nets', *ca* 170 cm long with two 1.6 m² openings) supplied by B&S Entomological Services. This is the same trap type used in Amazonia [18] and Costa Rica [16]. They were black with a white roof, and collected insects into approximately 80% ethanol (diluted from 96% undenatured ethanol supplied by Labtek Uganda Limited). We emptied traps at approximately two-week intervals.

We placed 16 traps in primary forest, 7 traps in disturbed forest, 9 traps in clearcut plantations and 2 traps outside the natural park in agricultural land (figure 1). We tried to cover as broad a range of habitats as possible: in primary forest, for example, we placed traps in swampy low ground and gravelly high ground, as well as the more common red-clay middle elevations.

Because an earlier sampling with randomized trap locations yielded few ichneumonids [25], we used more conventional methods for deciding on trap placement. We placed traps on or near likely insect flight paths, and tried to include as many 'special' habitats, such as fallen trees, as possible. Inevitably, this will have introduced some subjective bias to the results, but we feel this is an acceptable price for maximizing sample size and habitat coverage. The trap sites are described in greater detail in [52].

The total sampling effort was 857 samples, totalling 11 390.97 trap days or roughly 373 trap months. A further 19 samples (271.16 trap days ≈ 9 trap months, three rhyssines) were not included in the ecological analyses since they were unrepresentative of a normal catch: they include samples partly eaten by tree mice or trampled by elephants. All 34 traps were in use for most of the sampling year, with the exception of the very beginning (when not all traps were yet in place) and the last few months (when some traps wore out or were destroyed by hailstorms).

We processed the samples at the Zoological Museum of the University of Turku, Finland (ZMUT). We separated the ichneumonoid wasps (families Ichneumonidae and Braconidae), then pinned the rhyssines and sorted them into species (cf. [43]). The samples are deposited at ZMUT.

## 2.4. Statistical analyses

We classified our trap sites into five forest types: primary forest, swampy primary forest, disturbed forest, clearcut plantation and farmland. These are mainly based on the successional status of the forest (figure 1). However, we separated swamps from other primary forest because the vegetation around swamp traps differed from that of primary forest (NMDS ordination [52]). We also reclassified two trap sites (R93T1 and R93T2) into disturbed and primary forest, respectively, due to their vegetation resembling forest more than clearcut plantation [52].

To analyse how factors such as rainfall and forest type affect the number of rhyssines caught, we fitted a generalized linear model to our data. We modelled the effect of four ecological variables on expected rhyssine catches. For each species, we calculated the expected number of individuals in a sample as

$$\text{Catch} = \text{days} \cdot \exp(a_1 + a_2 \cdot \text{date} + a_3 \cdot \text{rain} + a_4 \cdot \text{deadwood} + a_{\text{foresttype}}), \tag{2.1}$$

where *days* is the number of days the sample was collected and was included as an offset, *date* is the number of days from 1 September 2014 00:00 to the middle of the sampling period, *rain* is the average

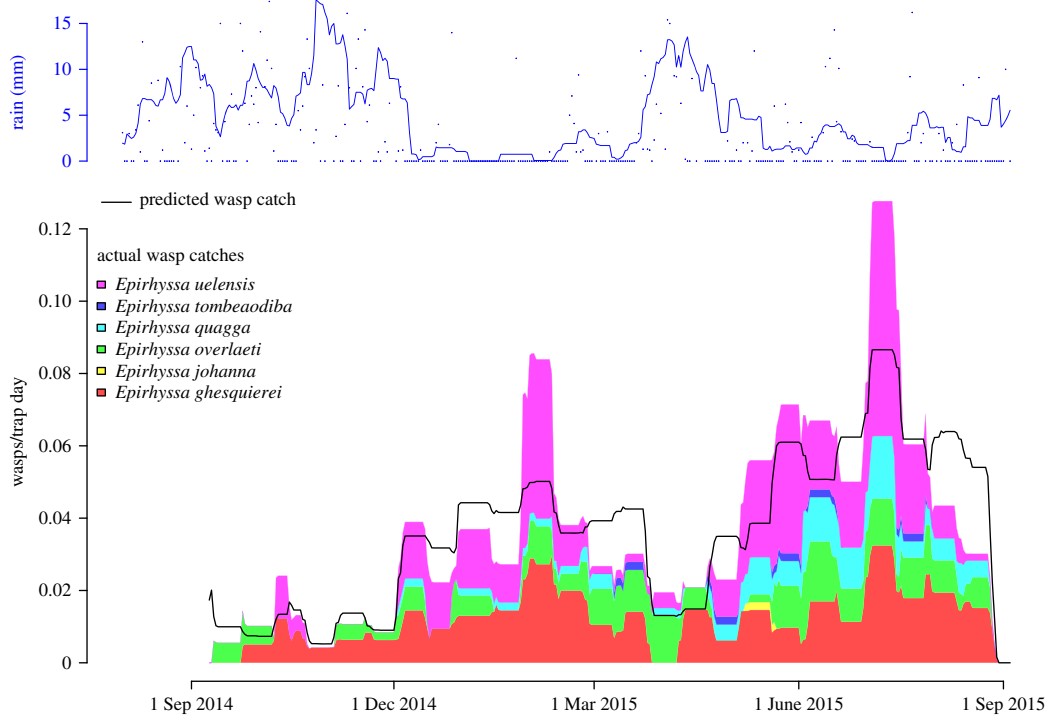

**Figure 2.** Number of rhyssines caught during the sampling year September 2014–September 2015, compared to the predicted catch and daily rainfall. Less individuals were caught when it rained, especially during the two rainy seasons, and catches increased with time. The catches of each wasp species are stacked (colour coded by species, as individuals per day per trap). The black line shows the total catch predicted by a model taking into account the date, rainfall, amount of decaying wood and forest type. Rainfall has been smoothed by taking 15 day averages (dots show actual rainfall), catches are averaged over the sampling period which for most samples was approximately 14 days. Catches after 4 September 2015 are not shown since no weather data were available then.

daily rainfall (in mm) during the time the sample was collected, *dead wood* is the amount of decaying wood near the trap and *forest type* is one of 'primary', 'swamp', 'disturbed', 'clearcut' or 'farm'. In calculating the amount of decaying wood, we only included clearly decaying wood (level of decay: 3) and gave the greatest weight to large pieces of wood near the trap (sum of $dbh^2$ distance$^{-1}$ in the two transects [52]).

We fitted this model to our data using the package mvabund of the R software [54,55]. We assumed negative binomial errors, and checked that they fit our data by plotting the residuals. To test the significance of each ecological variable, we calculated likelihood ratios for each species and compared them to a null distribution (acquired by resampling our trap sites 999 times). We tested the overall significance of each ecological variable (i.e. for the subfamily as a whole; sum of likelihood ratios of the species) by the same method. For the *forest type* variable, we calculated these likelihood ratios pairwise between the forest types. We did not adjust the probabilities for multiple testing since doing so would have introduced an unacceptable level of false negatives (type II errors); it should thus be noted that the results may contain false positives (type I errors).

All analyses were carried out in the R software, v. 3.4.0 [54]. We also used the R packages mvabund [55] and vegan (for NMDS ordination, [56]). The complete analyses and data are available online [52,57].

## 3. Results

We caught a total of 444 rhyssine individuals that were usable in ecological analyses. They belonged to six species, with sample sizes ranging from 160 individuals (*Epirhyssa uelensis*) to a single individual (*E. johanna*). Two species had less than 10 individuals.

Four ecological factors affected the number of rhyssines caught in traps. Catches increased later in the sampling year, and decreased in rainy weather (overall $p < 0.05$; figure 2). Traps also caught more rhyssines when surrounded by large amounts of decaying wood (overall $p < 0.05$; figure 3). Traps in

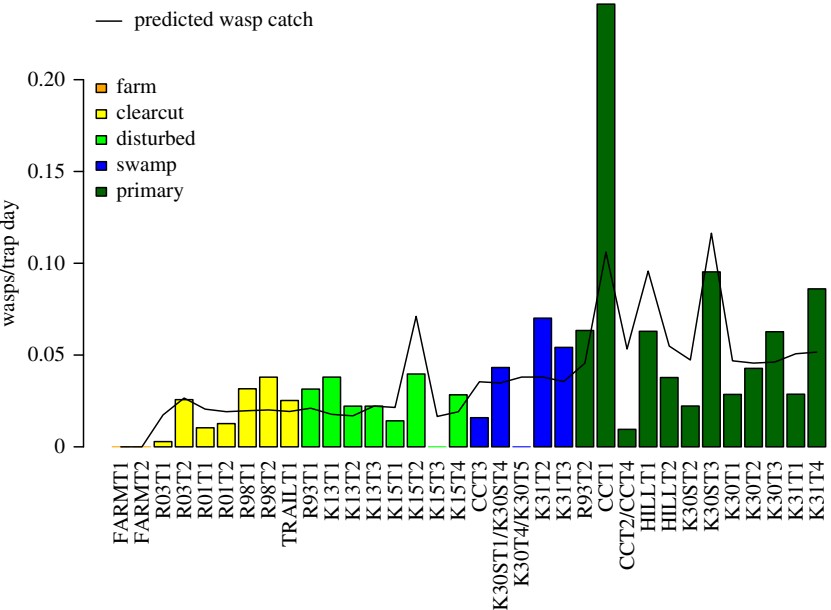

**Figure 3.** Number of rhyssines caught in 34 traps (as individuals per trap per day). Catches were highest in the primary forest trap CCT1 and lowest in farmland. The black line shows the catch predicted by a model taking into account the date, rainfall, amount of decaying wood and forest type. Catches after 4 September 2015 were not included in the model (due to lack of weather data) and are not shown.

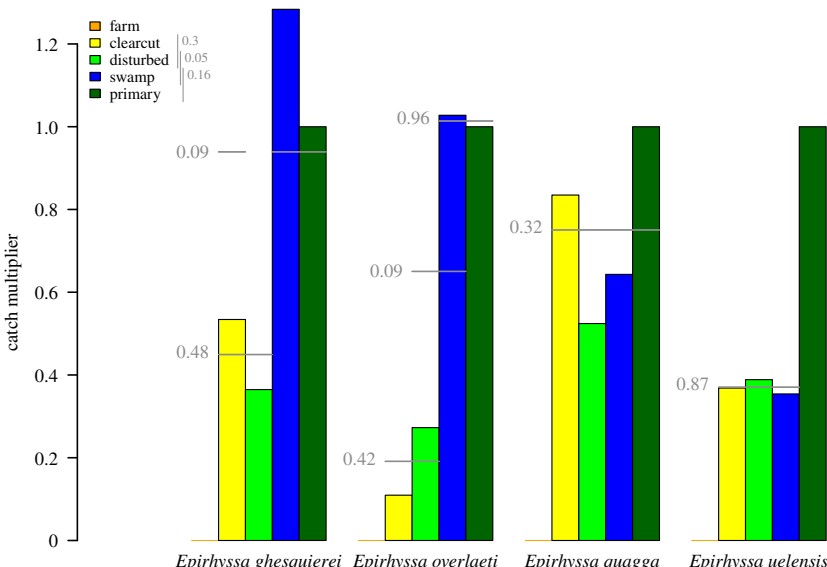

**Figure 4.** Effect of forest successional status on rhyssine catches. The species generally show a gradient from primary forest (catch size standardized to 1) down to farmland. Catch sizes are based on a model which also takes into account the date, rainfall and amount of decaying wood. Forest types that do not differ significantly ($p \geq 0.05$) are joined by a grey line, with the probability of getting the largest observed difference by chance displayed to the left. For the subfamily as a whole, clearcut plantation and disturbed forest did not differ from each other, nor did swamp and primary forest (grey lines in legend). The difference between disturbed forest and swamp was also insignificant ($p = 0.05$). We calculated significances by comparing the relative catches of the forest types to a null distribution (acquired by resampling the trap sites 999 times), and did not adjust significances for multiple testing. Two species are not displayed due to too low sample size ($n < 10$).

primary forest and swamp caught the most rhyssines, followed by traps in disturbed forest and clearcut plantation and then by traps in farmland (overall $p < 0.05$; figure 4). However, the difference between swamp and disturbed forest was not quite significant ($p = 0.05$; figure 4).

## 3.1. Phenology and habitat use of individual species

The phenology and habitat use of the six species largely followed that of the subfamily as a whole. Catches were affected by the date, rainfall, amount of decaying wood and forest type (table 1 and figure 2; electronic supplementary material, figures S1–S6). There were some exceptions to the overall pattern, however, especially for the species for which we had a small sample size.

*Epirhyssa ghesquierei* Seyrig, 1937 ($n = 154$) followed the overall pattern with one exception. Catches in clearcut plantation were not significantly lower than in primary forest and swamp ($p = 0.09$; figure 4).

*Epirhyssa johanna* sp. nov. Hopkins (in preparation) ($n = 1$) consists of one female caught in clearcut plantation, towards the end of the rainy season. The sample size was too low for ecological analyses.

*Epirhyssa overlaeti* Seyrig, 1937 ($n = 78$) followed the overall pattern with two exceptions. It was less affected by rain than other species, and the effect of rainfall was not significant ($p = 0.31$; table 1). The difference between swamp and disturbed forest was not significant ($p = 0.09$; figure 4).

*Epirhyssa quagga* sp. nov. Hopkins *et al.* (in preparation) ($n = 45$) followed the overall pattern with two exceptions. It was more abundant near decaying wood, but the effect was not significant ($p = 0.08$; table 1). It was absent from farmland, but otherwise showed no significant differences in its distribution between forest types (figure 4).

*Epirhyssa tombeaodiba* Rousse & van Noort, 2014 ($n = 6$) was caught in different traps during the latter half of the sampling year (electronic supplementary material, figure S5). The sample size was too low for ecological analyses.

*Epirhyssa uelensis* Benoit, 1951 ($n = 160$) followed the overall pattern with one exception. It appeared to be less common in swamp than in other primary forest ($p = 0.03$; figure 4), due to traps near running water (CCT3, K30T4/K30T5, K31T3) catching nothing. This may, however, be a false positive since the overall difference for the subfamily as a whole was not significant ($p = 0.16$ swamp versus primary forest).

# 4. Discussion

## 4.1. Four ecological variables affected catches

The four ecological variables of date, rainfall, amount of decaying wood and forest type affected the number of rhyssine adults caught by Malaise traps. Although there was some variation between species, the overall pattern was for catches to be highest near decaying wood in primary forest. The best time to catch rhyssines was in dry weather at the end of the sampling year.

The greatly increased catches near decaying wood strongly suggest that African rhyssines, like their counterparts elsewhere [12], parasitize immature stages of wood-boring endopterygote insects. Our Ugandan catches could be more than tripled by the presence of decaying wood (e.g. *Epirhyssa overlaeti*, trap CCT1: $\exp(4.42 * 0.27) = 3.3$). The hosts are presumably wood-boring beetle larvae, since there are probably very few wood-boring sawflies in the Afrotropics [58] and we cannot recall seeing any in our samples. Some of the observed variation in trap catches may be explained by our only measuring the amount of decaying wood, instead of actual host densities. Trap CCT1, for example, caught more than twice the number of rhyssines predicted by our model (figure 3). The log next to it (a fallen *Fagara macrophylla* (Oliv.) Engl. in a humid microclimate) attracted a conspicuous number of other decaying wood taxa such as long-ovipositored braconids and stilt-legged flies (Micropezidae), and seems to have been exceptionally suitable habitat for the hosts of rhyssines. Future sampling could benefit from also sampling wood-boring species with, for example, intercept traps.

African rhyssines appear to prefer undisturbed forest. Even after taking decaying wood into account, we found a successional gradient in catches: primary forest and swamp gave the highest catches, disturbed forest and clearcut plantation less, and farmland outside the national park yielded no rhyssines (figure 4). This has clear conservation implications: without the protection offered by the national park, all six species would presumably be locally extinct. It is unclear why forest type has such a strong effect, or rather why the effect persists after accounting for differences in the amount of decaying wood. We suspect that the wood of different forest types is not equivalent: factors such as microclimate, decay rate and tree species could all have an effect on what the host densities are in a given volume of wood. Factors unrelated to host availability, such as the humidity and shelter available to adults, may also play a part.

Ugandan rhyssines are caught in greater numbers when it does not rain (figure 2). This is the exact opposite of what little has earlier been observed in the tropics. In Costa Rica Pimplinae + Rhyssinae

**Table 1.** Coefficients of a model that predicts how the date, rainfall, amount of decaying wood and forest type affect rhyssine catches. Date and decaying wood and forest type affect rhyssine catches (positive coefficients), rain decreased catches (negative coefficients) and primary forest or swamp had the highest catches (highest coefficients). Significant coefficients ($p < 0.05$) in bold, except for forest type whose significances are displayed separately (figure 4). Two species are not displayed due to too low sample size ($n < 10$). The predicted daily catch for species sp is: $\mathrm{Catch_{sp}} = \exp(a_{sp,1} + a_{sp,2} \cdot \mathrm{date} + a_{sp,3} \cdot \mathrm{rain} + a_{sp,4} \cdot \mathrm{deadwood} + a_{sp,foresttype})$.

| species | intercept | date (days) | rain (mm d$^{-1}$) | dead wood (cm$^2$ m$^{-1}$) | forest type (primary) | forest type (swamp) | forest type (disturbed) | forest type (clearcut) | forest type (farm) |
|---|---|---|---|---|---|---|---|---|---|
| Epirhyssa ghesquierei | **−4.47** | **0.00197** | **−0.0831** | **3.57** | 0 | 0.2500 | −1.010 | −0.627 | −12.9 |
| Epirhyssa overlaeti | **−5.56** | **0.00360** | −0.0366 | **4.42** | 0 | 0.0273 | −1.300 | −2.210 | −12.3 |
| Epirhyssa quagga | **−7.38** | **0.00980** | **−0.1770** | 3.82 | 0 | −0.4410 | −0.646 | −0.181 | −10.8 |
| Epirhyssa uelensis | **−4.17** | **0.00543** | **−0.3740** | **2.31** | 0 | −1.0400 | −0.945 | −0.999 | −12.4 |

peaked at the end of the rainy season [16]. So did ichneumonids as a whole in a garden in Uganda [39]. Ichneumonids in moist forest in Panama also peaked in the rainy season [28]. The most likely explanation for the discrepancy between earlier results (peak in rainy season) and our results (less rhyssines when it rains) is that ours are the first results for the subfamily Rhyssinae alone. Earlier work has lumped Rhyssinae with other ichneumonid subfamilies, which may well have masked the subfamily's phenology. Our results indicate that adult rhyssines are either more active or more abundant—or probably a combination of both—when it is not raining. Their phenology may differ from that of other subfamilies due to their biology. Rhyssine hosts live inside decaying wood instead of feeding as free larvae on green vegetation, so rainfall probably has little effect on host densities. In the absence of any variation in host availability, adult rhyssines can be expected to fly when it is safest to do so, in clear weather. As well as being less active when it rains, it is plausible that adult rhyssine abundances are higher during the dry seasons, since the whole point of the adult phase is to search for hosts or mates, and rain interferes with this by making flying dangerous. It should also be noted that the undisturbed forest that the rhyssines mainly inhabit remains quite moist even during the dry seasons.

The two spikes in rhyssine catches in early February and July (figure 2) strongly suggest that Ugandan rhyssines peak in abundance then. These spikes occurred in the middle of the dry seasons and appear to represent actual increased abundances rather than just increased flight activity. This is because catches at the time were much higher than what our model predicted (figure 2), and our model already accounted for the normal effect of rainfall on rhyssine catches. The obvious interpretation is that Ugandan rhyssines preferentially emerge twice a year, in the middle of the dry season. This makes biological sense given that flight conditions are best then, and supports our hypothesis that seasonality has little effect on host availability for tropical rhyssines. More speculatively, the width of the spikes can be used to infer the adult lifespan. Both spikes in abundance lasted about two weeks (or less, traps were emptied at two-week intervals). The average lifespan of an adult Ugandan rhyssine is thus likely to be less than two weeks. This is much shorter than the 46 days or more measured for *Rhyssa persuasoria* (L.) when reared in colder climates [59], but not unrealistically short given that life expectancies may decrease with increased temperatures [60] and adult lifespans of about a week have been observed in other subfamilies (e.g. [60]). Predation, which is at its strongest in the tropics for at least some insect taxa [61], may also lead to shorter lifespans than what are achieved in rearings.

Overall catches rose steadily over the sampling year (figure 2). Catches were several times larger in August 2015 than they were in September 2014—at its most extreme, this pattern predicts over a 30-fold rise (for *Epirhyssa quagga*, table 1). We are unable to explain this rise, nor do we know if it occurs regularly every year or if we observed part of a longer trend. Factors such as the exceptionally strong 2015–2016 El Niño [62] and the generally less extreme weather of the latter two seasons we sampled (March–August) could have an effect, but this seems unlikely since we already included daily rainfall in our model. We are confident, however, that the rise reflects a genuine increase in abundances rather than more efficient trapping.

## 4.2. Extensive long-term sampling yielded ecological results

Our study followed the lead of earlier extensive sampling in Costa Rica [16] and Amazonia [18,26] in sampling tropical ichneumonids with large numbers of Malaise traps for a whole year. This gave us the sample size, as well as the coverage in time and space, that we needed to find out when and where rhyssine species fly.

Insufficient sample size has been the main reason why almost nothing was known about the ecology of Afrotropical rhyssines before this study: 30 published specimens for the whole of sub-Saharan Africa [42] are simply not enough. Our sample size of 444 individuals clearly demonstrates that extensive, long-term sampling can give the kind of sample sizes that are needed for ecological analyses. However, it should be noted that such sampling is no guarantee of success: our earlier survey at the same site caught no rhyssines (231 trap months with smaller traps and a slightly different sampling design [25]).

As well as giving a large sample size, extensive sampling also gave us a good coverage of different habitats and seasons. This allowed us to draw much more extensive conclusions on rhyssine habitat use and phenology than what has been possible before. It also contrasts with most earlier Afrotropical sampling, which tends to have been short term (e.g. [35]) or have involved only a few traps [39]. However, not even our sampling design gave complete coverage: we know of several habitats such as papyrus swamp and the canopy that we did not sample. Drawing more extensive conclusions would also benefit from an even longer sampling period than a year.

Our success in getting ecological data from the Afrotropics suggests using the same approach for other extensive sampling programmes. The extensive Amazonian (e.g. [18]) and Costa Rican (e.g. [16]) sampling programmes have up to now mainly yielded taxonomic results, out of necessity given the huge sample sizes and numerous newly discovered species. The ecology of the species has received less attention (but see [28]). Since our sampling designs have been almost identical and background data such as weather and vegetation are either obtainable or have already been collected (e.g. Amazonian vegetation data: [9]), we feel that it could be worthwhile revisiting the data of these sampling programmes. We are currently revisiting the Amazonian data, with the intention of getting comparable ecological information on the Amazonian rhyssines.

Our long-term aim is to describe how ichneumonid species are distributed on our planet, and to explain these distributions. Ecological data given by extensive sampling would be valuable in furthering this goal. Comparing the species richness of different sites, in particular, is much easier if we know what habitats and seasons have been sampled. We recommend the use of model-based methods (such as we have used here) in future sampling programmes, since models allow us to predict the catch sizes of a trap in an arbitrary habitat and weather. This would allow for easy comparison of sites irrespective of what habitats have been sampled, and potentially even help in explaining any observed differences in species richness.

## 4.3. Conclusions

Malaise trapping can reveal a lot more about tropical ichneumonids than just listing species, provided the ichneumonids are sampled for a long enough time and with enough traps. We paint a picture of Afrotropical rhyssines which avoid flying when it rains, presumably prey on the wood-boring larvae of decaying logs, and mainly live in natural forest. Our study also highlights how poorly we know the tropical ichneumonids. Further tropical sampling, and revisiting earlier results, will offer a wealth of useful data on the ecology of this family.

Ethics. The required research and export permits were issued by the Uganda National Council of Science and Technology (NS 504) and the Uganda Wildlife Authority. We minimized the harm to insect populations by using traps that collect a very small portion of flying (adult) insects, and by spacing the traps apart. We minimized the harm to individual insects by killing them quickly in 80% ethanol.

Data accessibility. The analyses and data are available online in two datasets. The first of these contains the code and data used in the statistical analyses, including data on the rhyssine wasps ([57], https://doi.org/10.5281/zenodo.2554871). The second contains background data on the Malaise trapping, such as trap coordinates, vegetation and weather data, and photographs of each trap site ([52], https://doi.org/10.5281/zenodo.2225643).

Authors' contributions. T.H. participated in designing the Malaise trapping, carried out the fieldwork, participated in processing the material, performed the analyses and drafted the manuscript; H.R. participated in designing the Malaise trapping, supervised the fieldwork and helped draft the manuscript; I.E.S. participated in designing the Malaise trapping, supervised the fieldwork, supervised the processing of the material and helped draft the manuscript. All authors gave final approval for publication.

Competing interests. We have no competing interests.

Funding. This study was funded by the Finnish Cultural Foundation, Oskar Öflunds Stiftelse, the Helsinki Entomology Society and Waldemar von Frenckells stiftelse (grants to T.H.).

Acknowledgements. Isaiah Mwesige helped maintain the Malaise traps and ably carried out a whole range of other fieldwork. Richard Sabiiti identified the plant species and helped gather the vegetation data. Yosinta Tumusiime gathered the weather data. Our field research was supported by the staff of the Makerere University Biological Field Station. Countless people helped process the samples, including the staff of the Zoological Museum of the University of Turku, students of the university and school pupils from throughout the Turku region. These contributions are gratefully acknowledged.

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
