## [Reviewer comments · Royal Society Open Science]

Review History

RSOS-190913.R0 (Original submission)

Review form: Reviewer 1

Is the manuscript scientifically sound in its present form?

Yes

Are the interpretations and conclusions justified by the results?

No

Is the language acceptable?

Yes

Is it clear how to access all supporting data?

Yes

Do you have any ethical concerns with this paper?

No

Have you any concerns about statistical analyses in this paper?

Yes

Recommendation?

Accept with minor revision (please list in comments)

Comments to the Author(s)

Please see the attached review (Appendix A). I liked the paper overall but had some minor problems with the post-hoc comparisons of catch among forest types, and their interpretation.

Review form: Reviewer 2**Is the manuscript scientifically sound in its present form?**

Yes

Are the interpretations and conclusions justified by the results?

Yes

Is the language acceptable?

Yes

Is it clear how to access all supporting data?

Yes

Do you have any ethical concerns with this paper?

No

Have you any concerns about statistical analyses in this paper?

No

Recommendation?

Accept with minor revision (please list in comments)

Comments to the Author(s)

Abstract. It is not necessary to mention the 'sister paper' in the abstract; this is for the main section.

Abstract. The number of sites should be added

Abstract. the last line is abit strange, and very specific to Costa Rica/ Amazonia (based on senior authors previous work), but I feel the sentence could be re-worded and made more general, so it encompasses other studies and for other regions. Arguably greater ecological information for Ichneumonids is useful for other regions too!

Intro, p1,147 "relatively" species-poor. There are still plenty of species in the tropics

p2 line 12-13 on sampling. While not from the tropics there are some useful papers missing that could give support to statements; eg Saunders TE & Ward DF. 2018. Variation in the diversity and richness of parasitoid wasps based on sampling effort. PeerJ 6:e4642

<https://doi.org/10.7717/peerj.4642>; and Fraser SEM, Dytham C, Mayhew PJ. 2008. The effectiveness and optimal use of Malaise traps for monitoring parasitoid wasps. *Insect Conservation and Diversity* 1(1):22-31

p2 lines 33-47. Much of this paragraph reports on previous papers without a very clear/direct link to the current work. Suggest reduce length and focus Africa/Rhyssinae

p3 12-3 "surrounded by dissimilar vegetation also catching dissimilar ichneumonids.."; this is confusing to read, can you clarify meaning. Are you saying there is some habitat partitioning of ichneumonids?; where different habitats have different assemblages?

methods - can you clarify the number of sites. you say 9, but there is also 2 on farmland?

p6 line 45 (and others) = you are mentioning an unpublished species name here. Ideally you should wait until the taxonomic revision paper is published

p7 lines 1-18. For most species it is mentioned that "probably a false positive caused by low sample size". First, I think this should be mentioned in the discussion, rather than repeat this for each species in results. Second, given your year long sampling; how many more samples and specimens would be needed to be sure?; you cannot just keep sampling until you get a significant result; and Third (and most importantly), perhaps that these species are generalists and the type of habitat is not so important, therefore the result is correct.

p7 line 32. sorry but you have NOT clearly demonstrated rhyssines are parasitoids of wood-boring insects. You have strong support for this, based on your increased catch with decaying wood

Decision letter (RSOS-190913.R0)

19-Jun-2019

Dear Mr Hopkins

On behalf of the Editors, I am pleased to inform you that your Manuscript RSOS-190913 entitled "Extensive sampling reveals the phenology and habitat use of Afrotropical parasitoid wasps (Hymenoptera: Ichneumonidae: Rhyssinae)" has been accepted for publication in Royal Society Open Science subject to minor revision in accordance with the referee suggestions. Please find the referees' comments at the end of this email.

The reviewers and handling editors have recommended publication, but also suggest some minor revisions to your manuscript. Therefore, I invite you to respond to the comments and revise your manuscript.

- Ethics statement

- Data accessibility

<http://datadryad.org/submit?journalID=RSOS&manu=RSOS-190913>

- Competing interests

- Authors' contributions

- Acknowledgements

- Funding statement

Because the schedule for publication is very tight, it is a condition of publication that you submit the revised version of your manuscript before 28-Jun-2019. Please note that the revision deadline will expire at 00.00am on this date. If you do not think you will be able to meet this date please let me know immediately.

If your manuscript is newly submitted and subsequently accepted for publication, you will be asked to pay the article processing charge, unless you request a waiver and this is approved by Royal Society Publishing. You can find out more about the charges at

<http://rsos.royalsocietypublishing.org/page/charges>. Should you have any queries, please contact openscience@royalsociety.org.

on behalf of Kevin Padian (Subject Editor)
openscience@royalsociety.org

Reviewer comments to Author:
Reviewer: 1

Comments to the Author(s)
Please see the attached review. I liked the paper overall but had some minor problems with the post-hoc comparisons of catch among forest types, and their interpretation.

Reviewer: 2

Comments to the Author(s)
Abstract. It is not necessary to mention the 'sister paper' in the abstract; this is for the main section.

Abstract. The number of sites should be added

Abstract. the last line is abit strange, and very specific to Costa Rica/Amazonia (based on senior authors previous work), but I feel the sentence could be re-worded and made more general, so it encompasses other studies and for other regions. Arguably greater ecological information for Ichneumonids is useful for other regions too!

Intro, p1,147 "relatively" species-poor. There are still plenty of species in the tropics

p2 line 12-13 on sampling. While not from the tropics there are some useful papers missing that could give support to statements; eg Saunders TE & Ward DF. 2018. Variation in the diversity and richness of parasitoid wasps based on sampling effort. PeerJ 6:e4642
<https://doi.org/10.7717/peerj.4642>; and Fraser SEM, Dytham C, Mayhew PJ. 2008. The effectiveness and optimal use of Malaise traps for monitoring parasitoid wasps. Insect Conservation and Diversity 1(1):22-31

p2 lines 33-47. Much of this paragraph reports on previous papers without a very clear/direct link tot the current work. Suggest reduce length and focus Africa/Rhyssinae

p3 12-3 "surrounded by dissimilar vegetation also catching dissimilar ichneumonids.."; this is confusing to read, can you clarify meaning. Are you saying there is some habitat partitioning of ichneumonids?; where different habitats have different assemblages?

methods - can you clarify the number of sites. you say 9, but there is also 2 on farmland?

p6 line 45 (and others) = you are mentioning an unpublished species name here. Ideally you should wait until the taxonomic revision paper is published

p7 lines 1-18. For most species it is mentioned that "probably a false positive caused by low sample size". First, I think this should be mentioned in the discussion, rather than repeat this for each species in results. Second, given your year long sampling; how many more samples and specimens would be needed to be sure?; you cannot just keep sampling until you get a significant results; and Third (and most importantly), perhaps that these species are generalists and the type of habitat is not so important, therefore the result is correct.

p7 line 32. sorry but you have NOT clearly demonstrated rhyssines are parasitoids of wood-boring insects. You have strong support for this, based on your increased catch with decaying wood

Author's Response to Decision Letter for (RSOS-190913.R0)

See Appendix B.

RSOS-190913.R1 (Revision)

Review form: Reviewer 1

Is the manuscript scientifically sound in its present form?

Yes

Are the interpretations and conclusions justified by the results?

Yes

Is the language acceptable?

Yes

Do you have any ethical concerns with this paper?

No

Have you any concerns about statistical analyses in this paper?

No

Recommendation?

Accept with minor revision (please list in comments)

Comments to the Author(s)

The manuscript has been improved from the earlier version. My concerns about false positives have been adequately dealt with and explained. My only minor comment is one that i had

intended to make in the earlier review but i obviously forgot top do so. It concerns the method the authors used in Figure 4 to distinguish statistical differences in standardized catches of Rhyssine species in different habitat types. I think it would be clearer to readers if the authors used the much more conventional system of denoting differences between habitats using different letters above the bars. I expect they used this other system as it allows them to specify the actual probabilities, rather than just those that are less than 0.05. However, if they wish to stick with the way they have done it, it is not a major sticking point with me.

Review form: Reviewer 2

Is the manuscript scientifically sound in its present form?

Yes

Are the interpretations and conclusions justified by the results?

Yes

Is the language acceptable?

Yes

Do you have any ethical concerns with this paper?

No

Have you any concerns about statistical analyses in this paper?

No

Recommendation?

Accept as is

Comments to the Author(s)

The authors have addressed the concerns raised by both reviewers. I think the ms should be accepted.

Decision letter (RSOS-190913.R1)

23-Jul-2019

Dear Mr Hopkins:

On behalf of the Editors, I am pleased to inform you that your Manuscript RSOS-190913.R1 entitled "Extensive sampling reveals the phenology and habitat use of Afrotropical parasitoid wasps (Hymenoptera: Ichneumonidae: Rhyssinae)" has been accepted for publication in Royal Society Open Science subject to minor revision in accordance with the referee suggestions. Please find the referees' comments at the end of this email.

The reviewers and Subject Editor have recommended publication, but also suggest some minor revisions to your manuscript. Therefore, I invite you to respond to the comments and revise your manuscript.

- Ethics statement

- Data accessibility

<http://datadryad.org/submit?journalID=RSOS&manu=RSOS-190913.R1>

- Competing interests

- Authors' contributions

- Acknowledgements

- Funding statement

Please note that we cannot publish your manuscript without these end statements included. We have included a screenshot example of the end statements for reference. If you feel that a given

heading is not relevant to your paper, please nevertheless include the heading and explicitly state that it is not relevant to your work.

Because the schedule for publication is very tight, it is a condition of publication that you submit the revised version of your manuscript before 01-Aug-2019. Please note that the revision deadline will expire at 00.00am on this date. If you do not think you will be able to meet this date please let me know immediately.

on behalf of Prof Kevin Padian (Subject Editor)
openscience@royalsociety.org

Associate Editor Comments to Author:

The reviewers are largely satisfied the paper is ready for publication; however, as you'll see, one of the reviewers has a minor suggestion for the presentation of one of the figures. It appears this is not essential to change, but you might like to consider making this modification for clarity.

Reviewer comments to Author:

Reviewer: 2

Comments to the Author(s)

The authors have addressed the concerns raised by both reviewers. I think the ms should be accepted.

Reviewer: 1

Comments to the Author(s)

The manuscript has been improved from the earlier version. My concerns about false positives have been adequately dealt with and explained. My only minor comment is one that i had intended to make in the earlier review but i obviously forgot to do so. It concerns the method the authors used in Figure 4 to distinguish statistical differences in standardized catches of Rhyssine species in different habitat types. I think it would be clearer to readers if the authors used the much more conventional system of denoting differences between habitats using different letters above the bars. I expect they used this other system as it allows them to specify the actual probabilities, rather than just those that are less than 0.05. However, if they wish to stick with the way they have done it, it is not a major sticking point with me.

Author's Response to Decision Letter for (RSOS-190913.R1)

See Appendix C.

Decision letter (RSOS-190913.R2)

25-Jul-2019

Dear Mr Hopkins,

I am pleased to inform you that your manuscript entitled "Extensive sampling reveals the phenology and habitat use of Afrotropical parasitoid wasps (Hymenoptera: Ichneumonidae: Rhyssinae)" is now accepted for publication in Royal Society Open Science.

Kind regards,

on behalf of Kevin Padian (Subject Editor)
openscience@royalsociety.org

Appendix A

General comments

This manuscript documents an extensive 1-year survey of Ichneumonid wasps in the subfamily Rhyssinae in different habitats (primary forests, swampy primary forests, disturbed forests, clearcuts, and farmland) in Kibale National Park, Uganda. The authors monitored catches every 2 weeks in 34 Malaise traps, collecting a total of 444 specimens in six species, all in the genus *Epirhyssa*. They used a general linear model to test for the influence of forest type, mean daily rainfall, date, and amount of decaying wood near a trap, on catch of each rhyssine species, as well as the subfamily as a whole. They found that rhyssine catches were greater in dry weather, in primary forest sites with lots of decaying logs, and towards the end of the sampling year.

I think this manuscript makes a significant contribution to our meager knowledge of habitat associations of parasitic wasps in the tropics, and especially in tropical Africa. It also outlines the type of studies and analyses that are needed to increase our understanding of the ecology of parasitic wasps. I am not familiar with the R program mvabund but I watched the introductory video (<http://eco-stats.blogspot.com/2012/03/introducing-mvabund-package-and-why.html>) and it appears to be a very good way of analyzing data like those collected by the authors.

However, I had an issue with the author's somewhat liberal interpretation of the results of the statistical analyses, specifically regarding differences in catch among the different forest types. For example, on page 7, lines 12–13, the authors state “We decided to accept the possibility of false positive values, and thus did not adjust the probabilities for multiple testing.” This is a very nonconventional statement. Most authors make an effort to maintain the experiment-wise error rate (probability of making a type I error) at the conventional 5% level by using a Bonferroni correction or by using post-hoc means tests like the Ryan-Einot Gabriel-Welsh test or Tukey's. I think it would be more acceptable if the authors were to use a Bonferroni correction for the comparison-wise error rate ($0.05/10 = 0.005$) for the ten pair-wise comparisons among the five different forest types for each rhyssine species. It is also not clear to me exactly what test was used to compare catches among the different forest types. This should be stated in the caption to Figure 4 as well as in the text. Alternatively, the authors could dispense with the post-hoc comparisons of catches among forest types and simply report the overall results of the general linear model that showing the effect of forest type on catch of rhyssine species, and refer the reader to Figure 4 to see the relative abundance of each species in Malaise traps set in each forest type.

My other beef in regard to stats interpretation is when the authors state more than once: “...but we interpret this as a false negative caused by too low a sample size.” While it may be prudent to caution the reader that ignoring an apparent trend in the data based on a borderline P value may result in a type II error, I don't think you can have it both ways, i.e., accept an inflated p value for making a type 1 error (by not reducing the comparison-wise error rate), and then dismissing the results when they are near the inflated p value. This could all be avoided by doing away with the post-hoc tests and presenting just the general conclusions from the model. The authors should also acknowledge that one year of sampling is very likely insufficient to draw too many conclusions about habitat associations of rhyssines, although it is a good start. It would be good to see parallel sampling of wood boring species (e.g., with Fluon-treated Lindgren funnel traps or intercept traps) as well as their parasitoids at the same sites to look for associations between both guilds and habitat types, rainfall, etc.

Specific comments and editorial comments (page numbers as printed in upper margin of manuscript).

- p. 3, lines 10, 18 Here, and elsewhere “e.g.” should be followed by a comma, i.e., “e.g.,”
- p. 3, line 21 I suggest the “e.g.” and the comma in this sentence be deleted.
- p. 3, line 25 Suggest “We have ~~also~~ published the results of a previous sampling in Uganda, but ~~this sampling~~ **that study** caught too few...”
- p. 3, line 44 “...different species are to be found, ~~with e.g.~~ e.g., with subfamilies such as...”
- p. 3, line 45 delete “also”
- p. 4, lines 12–13 Suggest joining these sentences with a semi-colon” “In general, rhyssines are held to be parasitoids of wood-boring sawfly or beetle larvae [12,16] that prefer humid lowland forests [16]; dry periods and arid areas ~~seem not to be~~ **appear unsuitable** [38,39].”
- p. 4, lines 18–19 Suggest “**Prior to the present study,** ~~For African rhyssines, the situation before this work is easily summarised:~~ only 30 *individuals of African rhyssines* were known from the whole of Sub-Saharan Africa ([40], though some further specimens may exist in undetermined museum collections).”
- p. 4, line 21 “Ecological analyses are impossible with such sample sizes, and, ~~have~~ as far as we know, **have** not been attempted.”
- p. 4, lines 25–28 Replace the dash before “i.e., with a comma, and follow with a comma, i.e., “Here, we describe the ecology of this subfamily. Specifically, we asked how the rhyssine species were distributed in space and time, **i.e.,** —i.e. how ecological variables such as weather and habitat type affected the number of rhyssines caught in traps.”

This sentence suggests the authors have evidence of cause and effect between weather, habitat type and the numbers of rhyssines captured in traps. I think at this point it would be more prudent to state the **associations** the authors observed between apparent rhyssine abundance (as measured by trap catches) and both weather and habitat type. Many factors may affect the capture of rhyssines in traps in addition to their abundance near the trap, e.g., trap height, trap exposure (i.e., openness to wind and flight corridors), surrounding vegetation, etc.

- p. 5, line 23 Insert comma after “i.e.”
- p. 6, line 6 Ethanol is known to be emitted in larger quantities of trees under stress, and is also a synergist for attraction of cerambycid beetles when combined with their aggregation pheromones. Therefore, it is conceivable that some rhyssine parasitoids of cerambycid larvae may be attracted to ethanol, due to its association with stressed trees and potential host presence.
- p. 7, line 8 The negative binomial is often the best fit for count data but not always – sometimes a Poisson or even a Gaussian distribution fit count data better than the negative binomial (e.g., I have found that species richness per trap is often fit well by the normal distribution). Did you test other distributions and compare the goodness of fit with something like Aikake’s Information Criterion?

Appendix B

Response to reviewers (RSOS-190913)

We are grateful to the two reviewers for their comments and suggestions, which we have incorporated into the revised manuscript. We address the issues raised by the reviewers in this response. Revisions to the paper have been made as tracked changes.

The main need for revision that the reviewers identified was in how we compared different forest types and interpreted these results. Other suggested changes were relatively minor.

Reviewer 1

However, I had an issue with the author's somewhat liberal interpretation of the results of the statistical analyses, specifically regarding differences in catch among the different forest types. For example, on page 7, lines 12–13, the authors state "We decided to accept the possibility of false positive values, and thus did not adjust the probabilities for multiple testing." This is a very nonconventional statement. Most authors make an effort to maintain the experiment-wise error rate (probability of making a type I error) at the conventional 5% level by using a Bonferroni correction or by using post-hoc means tests like the Ryan-Einot Gabriel-Welsh test or Tukey's. I think it would be more acceptable if the authors were to use a Bonferroni correction for the comparison-wise error rate ($0.05/10 = 0.005$) for the ten pair-wise comparisons among the five different forest types for each rhyssine species.

We have rewritten this sentence to make it clearer why we did not correct for multiple testing. The issue of whether or not to do Bonferroni corrections (or the equivalent) is much discussed. In our opinion, not doing so is justified in our paper.

The purpose of methods like the Bonferroni correction is to entirely eliminate false positives (type I errors). Their answer to the question "How many false positives are acceptable?" is zero, irrespective of the number of tests made.

Our analyses involve a large number of tests (we test six species and four variables, one of which is split into ten pairwise comparisons). If left unadjusted, our results will likely include one or more false positives. Applying e.g. a Bonferroni correction, on the other hand, would result in no false positives ($p < 0.05$), but would introduce a vast number of false negatives.

In our opinion, the first of these options is more desirable in this paper. We prefer to accept a few false positives, and bear their presence in mind when interpreting the results, rather than (potentially) obtain no significant results whatsoever.

We have rewritten the sentence to include the above argument, and also to highlight the fact that the results may contain false positives. We have also toned down our interpretations of these results.

It is also not clear to me exactly what test was used to compare catches among the different forest types. This should be stated in the caption to Figure 4 as well as in the text.

We have made this clearer in the methods and the caption to Figure 4. The test was the same as for the other ecological variables (likelihood ratios compared to a null distribution obtained by resampling). For the forest type, we calculated likelihood ratios pairwise between the forest types.

Alternatively, the authors could dispense with the post-hoc comparisons of catches among forest types and simply report the overall results of the general linear model that showing the effect of forest type on catch of rhyssine species, and refer the reader to Figure 4 to see the relative abundance of each species in Malaise traps set in each forest type.

We seriously considered this alternative, since it would make the analyses much simpler and easier to describe to the reader. However, it would also have led to discarding useful information on the habitat use of the individual species.

My other beef in regard to stats interpretation is when the authors state more than once: “...but we interpret this as a false negative caused by too low a sample size.” While it may be prudent to caution the reader that ignoring an apparent trend in the data based on a borderline P value may result in a type II error; I don’t think you can have it both ways, i.e., accept an inflated p value for making a type I error (by not reducing the comparison-wise error rate), and then dismissing the results when they are near the inflated p value.

Both reviewers raised this issue. We agree, and have deleted these statements.

The purpose of these statements was to highlight results which we had reason to believe are false negatives. For example, the insignificant *E. overlaeti* difference between disturbed and swampy forest ($p=0.09$, Fig. 4) is likely a false negative. This is because these two forest types respectively group with clearcut plantation ($p=0.42$) and primary forest ($p=0.96$), yet clearcut and primary differ ($p<0.05$). In other words, these pairwise results are mutually contradictory, and one of them must logically be a false positive or negative. The most likely false result is the $p=0.09$, near-significant result.

However, this is very hard to phrase clearly. There is also no guarantee that we have correctly identified which result is false. We thus decided to delete all such statements, and leave any interpretation of such results to the reader.

p. 3, lines 10, 18 Here, and elsewhere “e.g.” should be followed by a comma, i.e., “e.g.,”

p. 3, line 21 I suggest the “e.g.” and the comma in this sentence be deleted.

p. 3, line 25 Suggest “We have also published the results of a previous sampling in Uganda, but ~~this sampling~~ that study caught too few...”

p. 3, line 44 “...different species are to be found, with e.g. e.g., with subfamilies such as...”

p. 3, line 45 delete “also”

p. 4, lines 12–13 Suggest joining these sentences with a semi-colon” “In general, rhyssines are held to be parasitoids of wood-boring sawfly or beetle larvae [12,16] that prefer humid lowland forests [16]; dry periods and arid areas seem not to be appear unsuitable [38,39].”

*p. 4, line 21 “Ecological analyses are impossible with such sample sizes, and, ~~have~~ as far as we know, **have not** been attempted.”*

p. 5, line 23 Insert comma after “i.e.”

We have adopted the above suggestions, with the exception of the comma after i.e. and e.g. Placing a comma after e.g. is a matter of style, with American English strongly favouring the comma and British English tending to omit it. We have consistently omitted the comma throughout the paper.

We have also corrected our misuse of "sampling" as a noun (*p. 3 line 25*) elsewhere in the text, specifically in section 4.2.

*p. 4, lines 18–19 Suggest “**Prior to the present study**, ~~For African rhyssines, the situation before this work is easily summarised: only 30 individuals of African rhyssines were known from the whole of Sub-Saharan Africa ([40], though some further specimens may exist in undetermined museum collections).~~”*

We have adopted this suggestion in modified form.

p. 4, lines 25–28 *Replace the dash before “i.e., with a comma, and follow with a comma, i.e., “Here, we describe the ecology of this subfamily. Specifically, we asked how the rhyssine species were distributed in space and time, i.e., —i.e. how ecological variables such as weather and habitat type affected the number of rhyssines caught in traps.”*

*This sentence suggests the authors have evidence of cause and effect between weather, habitat type and the numbers of rhyssines captured in traps. I think at this point it would be more prudent to state the **associations** the authors observed between apparent rhyssine abundance (as measured by trap catches) and both weather and habitat type. Many factors may affect the capture of rhyssines in traps in addition to their abundance near the trap, e.g., trap height, trap exposure (i.e., openness to wind and flight corridors), surrounding vegetation, etc.*

We have adopted this change (with the exception of the comma after "i.e."), and made it clear we are looking at associations between rhyssine catches and ecological factors, not direct cause and effect.

p. 6, line 6 *Ethanol is known to be emitted in larger quantities of trees under stress, and is also a synergist for attraction of cerambycid beetles when combined with their aggregation pheromones. Therefore, it is conceivable that some rhyssine parasitoids of cerambycid larvae may be attracted to ethanol, due to its association with stressed trees and potential host presence.*

This is an interesting point, and one we had overlooked! Any effect is likely small since the forest at our study site is full of fruit trees and other stronger sources of ethanol. But we did observe some taxa (mainly flies) attracted to our traps.

We feel discussing the intricacies of Malaise trapping at this level of detail goes outside the scope of our paper, but will definitely keep this point in mind for future work. We are e.g. planning to compare the data of different surveys, and this would be relevant there.

p. 7, line 8 *The negative binomial is often the best fit for count data but not always – sometimes a Poisson or even a Gaussian distribution fit count data better than the negative binomial (e.g., I have found that species richness per trap is often fit well by the normal distribution). Did you test other distributions and compare the goodness of fit with something like Aikaike’s Information Criterion?*

We have added a brief mention of this.

We visually assessed the fit of the Poisson and negative binomial distributions by plotting their residuals (function `plot.manyglm` in the R package `mvabund`). We felt these two distributions to be the most meaningful for count data.

Both distributions fit the data quite well. We decided on the negative binomial partly because of its good fit, but also because it is more likely than the Poisson to fit future data for other subfamilies. We intend to analyse the other Ugandan subfamilies using the same methods as in this paper, and feel that having the same distribution in each paper would make the analyses more consistent.

The authors should also acknowledge that one year of sampling is very likely insufficient to draw too many conclusions about habitat associations of rhyssines, although it is a good start. It would be good to see parallel sampling of wood boring species (e.g., with Fluon-treated Lindgren funnel traps or intercept traps) as well as their parasitoids at the same sites to look for associations between both guilds and habitat types, rainfall, etc.

We have added these to the discussion. Parallel sampling of wood-boring species would indeed be a very good next step!

Reviewer 2

Abstract. It is not necessary to mention the 'sister paper' in the abstract; this is for the main section.

We have removed the "sister paper" from the abstract.

Abstract. The number of sites should be added

We have made this change.

Abstract. the last line is abit strange, and very specific to Costa Rica/Amazonia (based on senior authors previous work), but I feel the sentence could be re-worded and made more general, so it encompasses other studies and for other regions. Arguably greater ecological information for Ichneumonids is useful for other regions too!

We have made the sentence more general by removing mention of Costa Rica/Amazonia.

However, in practice the sentence still largely refers to Costa Rica and Amazonia in the context of the tropics, since these are the only two tropical areas that have been sampled extensively enough.

Intro, p1,147 "relatively" species-poor. There are still plenty of species in the tropics

We have made this change.

*p2 line 12-13 on sampling. While not from the tropics there are some useful papers missing that could give support to statements; eg Saunders TE & Ward DF. 2018. Variation in the diversity and richness of parasitoid wasps based on sampling effort. PeerJ 6:e4642 <https://doi.org/10.7717/peerj.4642>; and Fraser SEM, Dytham C, Mayhew PJ. 2008. The effectiveness and optimal use of Malaise traps for monitoring parasitoid wasps. *Insect Conservation and Diversity* 1(1):22-31*

We thank the reviewer for suggesting these excellent papers, and have cited them as suggested. (This change does not show in tracked changes due to the citation software we use.)

p2 lines 33-47. Much of this paragraph reports on previous papers without a very clear/direct link tot the current work. Suggest reduce length and focus Africa/Rhyssinae

We have rewritten parts of the paragraph to make the link to this paper clearer, and deleted some less relevant details.

The purpose of this paragraph is to summarise what little was previously known of ichneumonid phenology and habitat use in the tropics. We feel it is worth giving at least some broader context (tropics/Ichneumonidae, not just Africa/Rhyssinae), especially since so little is known of African rhyssines.

p3 12-3 "surrounded by dissimilar vegetation also catching dissimilar ichneumonids.."; this is confusing to read, can you clarify meaning. Are you saying there is some habitat partitioning of ichneumonids?; where different habitats have different assemblages?

We have made this sentence clearer. There was indeed some habitat partitioning: ichneumonid assemblages were associated with the vegetation around traps.

methods - can you clarify the number of sites. you say 9, but there is also 2 on farmland?

We have added a mention of the farmland as a tenth "site" to make this clearer. There were 9 sites in the national park, and 2 traps in farmland outside the park.

p6 line 45 (and others) = you are mentioning an unpublished species name here. Ideally you should wait until the taxonomic revision paper is published

We agree that there would be a case for waiting until the two species names are published. However, we cite the current manuscript (especially the sampling methods) in our taxonomic paper, so there is also a case for getting this manuscript published first. Ideally we would publish both simultaneously, but getting two journals to synchronise their publication schedule is likely not feasible (although we must confess to having toyed with the idea..).

We feel this is not too much of a problem, since the taxonomy paper is almost certain to appear by the end of the year (likely sooner), and we are confident that the species names we have selected (*Epirhyssa quagga* and *E. johanna*) will not change.

p7 lines 1-18. For most species it is mentioned that "probably a false positive caused by low sample size". First, I think this should be mentioned in the discussion, rather than repeat this for each species in results. Second, given your year long sampling; how many more samples and specimens would be needed to be sure?; you cannot just keep sampling until you get a significant results; and Third (and most importantly), perhaps that these species are generalists and the type of habitat is not so important, therefore the result is correct.

Both reviewers raised this issue. We agree that these statements go beyond what is supported by the results, and have deleted them. See response to reviewer 1 for further details.

p7 line 32. sorry but you have NOT clearly demonstrated rhyssines are parasitoids of wood-boring insects. You have strong support for this, based on your increased catch with decaying wood

Quite right! We have corrected the sentence.

Appendix C

Response to Reviewer 1 (RSOS-190913)

We decided not to adopt the suggestion made by Reviewer 1 for the reasons below. The uploaded files are thus the same as previously.

Reviewer 1: The manuscript has been improved from the earlier version. My concerns about false positives have been adequately dealt with and explained. My only minor comment is one that i had intended to make in the earlier review but i obviously forgot to do so. It concerns the method the authors used in Figure 4 to distinguish statistical differences in standardized catches of Rhyssine species in different habitat types. I think it would be clearer to readers if the authors used the much more conventional system of denoting differences between habitats using different letters above the bars. I expect they used this other system as it allows them to specify the actual probabilities, rather than just those that are less than 0.05. However, if they wish to stick with the way they have done it, it is not a major sticking point with me.

We decided not to make this change. Letters above the bars are indeed more conventional, but in our opinion they are often unclear. We prefer to try our current system in the hope that it will be easier to interpret (once the reader is accustomed to the system, at least). As the reviewer pointed out, it also has the advantage of showing actual probabilities. This can help readers interpret figure 4 in more detail.